# Sparse Universal Transformer

**Shawn Tan**[1] [*]
tanjings@mila.quebec

**Yikang Shen**[2] [*]
yikang.shen@ibm.com

**Zhenfang Chen**[2]
zfchen@ibm.com

**Aaron Courville**[1]
courvila@iro.umontreal.ca

**Chuang Gan**[2]
chuangg@ibm.com

[1]Mila, University of Montreal

[2]MIT-IBM Watson AI Lab

## Abstract

The Universal Transformer (UT) is a variant of the Transformer that shares parameters across its layers. Empirical evidence shows that UTs have better compositional generalization than Vanilla Transformers (VTs) in formal language tasks. The parameter-sharing also affords it better parameter efficiency than VTs. Despite its many advantages, scaling UT parameters is much more compute and memory intensive than scaling up a VT. This paper proposes the Sparse Universal Transformer (SUT), which leverages Sparse Mixture of Experts (SMoE) and a new stick-breaking-based dynamic halting mechanism to reduce UT's computation complexity while retaining its parameter efficiency and generalization ability. Experiments show that SUT achieves the same performance as strong baseline models while only using half computation and parameters on WMT'14 and strong generalization results on formal language tasks (Logical inference and CFQ). The new halting mechanism also enables around 50% reduction in computation during inference with very little performance decrease on formal language tasks.

## 1 Introduction

Recent theoretical work has pointed out that finite-depth Transformers have an issue of expressibility that will result in failure to generalize (Hahn, 2020; Hao et al., 2022; Merrill et al., 2022; Liu et al., 2022). Delétang et al. (2022) ran several neural architectures on a suite of different synthetic languages generated from different levels of the Chomsky hierarchy and empirically confirmed these results, showing that VTs have difficulty generalizing to Regular languages. Universal Transformers (UTs; Dehghani et al. 2018) are Transformers that share parameters at every layer of the architecture. Csordás et al. (2021) performed several compositional generalization experiments on VTs and UTs

---
[*] Equal contribution

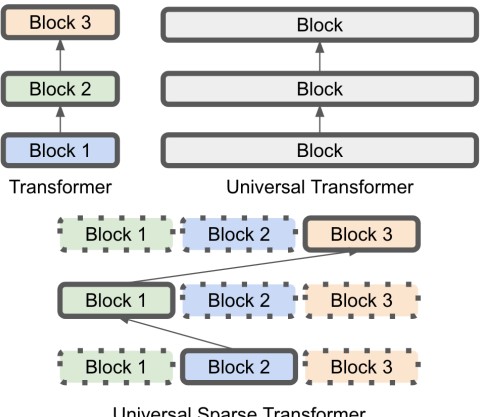

Figure 1: A VT has separate Transformer blocks for each layer, with different parameters. For a UT with the same number of parameters, the UT block will be ∼3 times the dimensions of each VT block. Running this block for 3 layers would then incur approximately 9 times the runtime memory. Using SMoEs can recover approximately the same computational cost as the VT.

along with absolute and relative position embeddings, and showed that UTs with relative positional embeddings performed better on these tasks.

However, the task of scaling UTs is challenging due to its computation complexity (Kaplan et al., 2020; Tay et al., 2022; Takase and Kiyono, 2021). Consider a VT with $P$ parameters for each layer and $L$ layers. Evaluating such a VT has computation complexity associated with the model size $LP$. A size-equivalent UT would have a UT block with $LP$ parameters and computation complexity of approximately $LP$ to run the block once. To run such a UT for equivalent $L$ layers would incur a complexity of $L^2 P$. This increased computation complexity directly translates to increased training and inference time. According to Takase and Kiyono (2021), UT requires two times the training time and far more GPU memory than VT in WMT English-German translation task.

Sparsely activated neural networks were intro-

duced to reduce the computation complexity of large models. These networks activate parts of the network conditioned on the input, computing only parts of the model, thereby disentangling the number of parameters from the computation complexity. This method allows for drastically increasing the number of parameters without proportionally increasing the computation complexity. Shazeer et al. (2017) introduced Sparse Mixture of Experts (SMoE), using the top-$k$ operator to allow for sparse computation of experts. This allows for replacing the FeedForword (FFD) layer in the Transformer with an ensemble of $E_{\text{ffd}}$ FFDs, but only $k$ FFDs (where $k < E$) would have to be evaluated, conditioned on the input. Zhang et al. (2022) then introduced the Mixture of Attention heads (MoA), which allows Transformers to replace its Multihead Attention (MHA) layer with an ensemble of $E_{\text{att}}$ attention heads and only activates $k$ heads condition on the input, further sparsifying the model.

This paper introduces the Sparse Universal Transformer (SUT), which applies the above sparse computation techniques to UT. Additionally, we replace the per-position halting mechanism in UT with a new stick-breaking formulation that has a probabilistic interpretation, allowing us to introduce an Adaptive Computation Time (ACT; Graves 2016) penalty to minimize layer use. It also provides an easy way to adjust the trade-off between the amount of computation and model performance during inference, further improving the efficiency of the SUT at inference time.

To demonstrate effective scaling, we perform experiments on WMT'14 English to German translation, showing that an SUT can achieve better performance for the same parameter count, while incurring less computation cost than an equivalent dense UT. Since the UT setting is a specific case of SUT, we show on the Compositional Freebase Questions (CFQ; Keysers et al. 2019) tasks that UTs have better compositional generalization properties, improving upon CFQ results from Csordás et al. (2021). Using the Logical Inference task (Bowman et al., 2015), we analyse the behaviour of our UT on length and compositional generalization. Finally, we show that the halting mechanism can be used to achieve further efficiency during inference time, and study the trade-off between efficiency and performance.

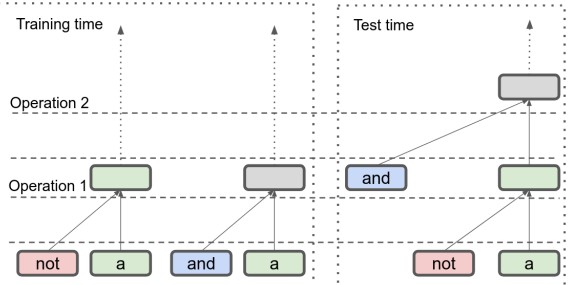

Figure 2: Example of the compositional generalization splits from (Shen et al., 2019). The combination of not and and are never seen in successive combination during training, and a VT may learn a shortcut that prevents generalisation during test.

## 2  Background & Related Work

**Overcoming VT limitations with UT**  Dziri et al. (2023) and Liu et al. (2022) find that Vanilla Transformers learn shortcuts for tasks that require multistep compositional operations, and fail to generalize on larger instances of the problem that require more steps. Theoretical results have also shown that Vanilla Transformers have limitations in what they can compute that support these findings (Hahn, 2020; Hao et al., 2022; Merrill et al., 2022). Universal Transformers (Dehghani et al., 2018) are Transformers with tied weights across all layers, and an additional halting mechanism to decide when to stop. In an ideal scenario of infinite layers (now that all layers have the same parameters) the UT, like the Neural GPU (Kaiser and Sutskever, 2015), is Turing-complete, which overcomes many of the abovementioned issues.

In practice, even with limited depth, UTs have exhibited properties that afford them better performance in compositional generalization tasks (Csordás et al., 2021). UTs allow operations learned in the Transformer during training to be depth-order invariant. If some operations during training are learned to be performed in a certain order, during test time, the UT could generalize to an unseen order of operations.

**Challenges with Scaling the UT**  Despite these compositional abilities, performance tends to decrease on real-world tasks when using UTs. AL-BERT (Lan et al., 2019) improved parameter efficiency by sharing parameters across layers. This was motivated by an observation that Transformers tend to learn to perform similar operations in the layers, and that sharing these parameters would

reduce this redundancy[1]. However, the authors observe a dip in performance when sharing parameters, contrary to Dehghani et al. (2018).

Could the issue be one of model capacity? Experiments with ALBERT show that scaling up ALBERT can outperform the BERT baseline, even on real-world tasks (Lan et al., 2019). Kaplan et al. (2020) also show that a shared-parameter Transformer has better scaling properties in terms of parameter-to-performance, but poorer properties in terms of computation-to-performance, since parameter count causes the computation to increase. Tay et al. (2022) scale up different sequence models, and remark on difficulties with scaling up UTs, limiting the experiments they can perform on UT. Takase and Kiyono (2021) outline several strategies of scaling up shared-parameter Transformers to deal with these issues by using different parameter-sharing schemes.

Our experiments show that SMoE techniques can be applied successfully to the UT to scale it up, achieving the UT's parameter efficiency while not incurring the same computation costs. We also perform experiments that support the compositional generalization claims of prior work, and provide better baselines for those tasks.

## 3  Method

Like UT, we reuse the same SUT block for every layer of the Transformer. Within each SUT block, we use SMoEs to achieve sparsity for the feed-forward network (FFD) and attention heads separately. We use the Mutual Information Maximization loss proposed in Chen et al. (2022) and modified for unsupervised tasks in Shen et al. (2023). Finally, we propose a stick-breaking process formulation of dynamic halting, which affects how the attention mechanism works in the SUT, and the Adaptive Computation Time (ACT) auxiliary loss we use to minimize the number of layers used.

### 3.1  Sparse Mixture of Experts

A Mixture of Experts module consists of $E$ sub-modules $f_1, \ldots, f_E$. There is also a gating network, which we will denote by $g(e \mid \mathbf{h})$ – for any input $\mathbf{h}$ to the MoE module, the gating network would predict a distribution over the $E$ experts. When $k < E$, we refer to this as a Sparse Mixture of Experts (SMoE), and $g(e \mid \mathbf{h}) > 0$ for only $k$ ex-

perts, while maintaining that $\sum_e^E g(e \mid \mathbf{h}) = 1$. The final output of the SMoE is then given by $y = \sum_{e=1}^E g(e \mid \mathbf{h}) \cdot f_e(\mathbf{h})$, where $g(e \mid \mathbf{h}) = 0$, $f_e(\mathbf{h})$ will not need to be evaluated, reducing computation cost during training and inference. We replace the Feed-forward layer (FFD) in the Transformer block with a mixture of FFDs. Each Mixture of FFD can be described with a 3-tuple, $(E, k, D)$: $E$ experts, $k$ for the number of experts to be used in the top-$k$ operation, and $D$ for the dimension of the hidden layer for each FFD expert. For the attention layer, we use the Mixture of Multi-head Attention (MoMHA) proposed by Zhang et al. (2022) and Shen et al. (2023). Each MoMHA module can be described by a 5-tuple, $(E, k, H, D, W)$, with $E$ representing the number of experts, $K$ representing the parameter $k$ in a top-$k$ operation, $H$ representing the number of attention heads per expert, and $D$ for the dimensions per head. Like in MoA, MoMHA maintains only a single set of $H$ key-value projections shared among the experts, while there are $E$ query projections of $H$ heads each. $W$ represents the relative position embedding window size, parameterizing $2W + 1$ embeddings for $W$ positions before and after the present position. Figure 3 (*Left*) shows the schematic of a SUT block.

This technique has been used to reduce computation costs both during training and inference time for large models.

**Mutual Information Maximisation**  Like other models that rely on conditional activation, auxiliary losses are needed in order to aid learning a module that decides which experts are activated, and to ensure that all experts are used, balancing the load for processing. For this, we use the Mutual Information Maximization introduced in Chen et al. (2022) for the auxiliary loss (to be maximised):

$$\mathcal{L}_{\text{MIM}} = \underbrace{\sum_{e=1}^E g(e) \log g(e)}_{-H(e)}$$

$$\underbrace{- \frac{1}{|\mathcal{X}|} \sum_{\mathbf{h} \in \mathcal{X}} \sum_{e=1}^E g(e \mid \mathbf{h}) \log g(e \mid \mathbf{h})}_{H(e \mid \mathbf{h})}, \quad (1)$$

where,

$$g(e) = \frac{1}{|\mathcal{X}|} \sum_{\mathbf{h} \in \mathcal{X}} g(e \mid \mathbf{h})$$

[1]https://ai.googleblog.com/2019/12/albert-lite-bert-for-self-supervised.html

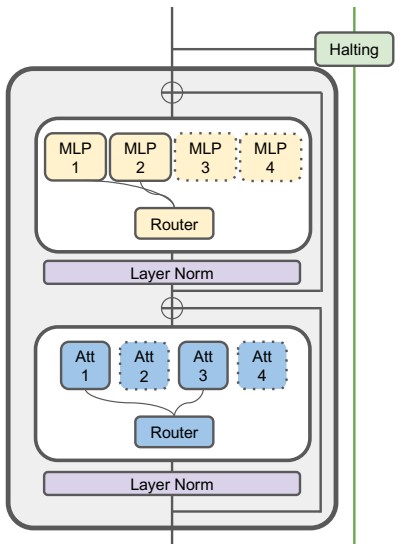 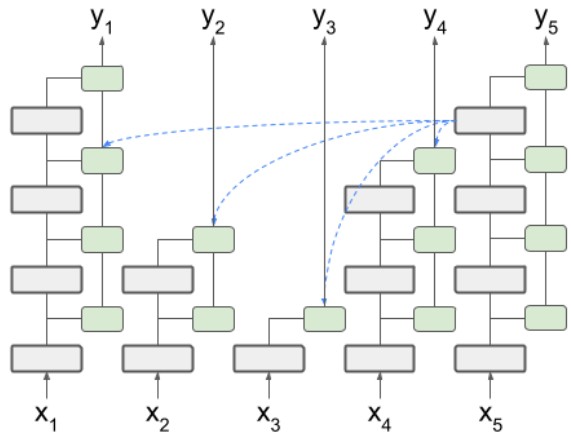

Figure 3: *Left*: Schematic of a SUT block.   *Right*: While the input of each SUT block is the output of the previous layer, the attention mechanism attends to the halted state of the timestep. When the halting probability exceeds $\alpha_{\text{thresh}}$, the hidden state is simply copied. Finally, the halted state is used as the output of the SUT.

. Specifically, we use the unsupervised version proposed by Shen et al. (2023) that assumes a uniform distribution over all tokens and layers, resulting in the following auxiliary objective. In the SUT setting, the gating network is used $|\mathcal{X}| = L \cdot T$ times, where $L$ is the number of layers, and $T$ is the number of timesteps.

Intuitively, the entropy term increases the entropy of the marginal probability of the gating network predictions, which at its maximum means that the weight for each gating network across the entire minibatch is uniform. The conditional entropy term decreases the conditional entropy, which causes the prediction of the gating network to be sharp, and also penalizes the uniform distribution solution for the gating network.

## 3.2 Stick-breaking Dynamic Halting

There have been several methods for imbuing models with the ability to make a prediction without having to use all layers of the model (Graves, 2016; Tan and Sim, 2016; Dehghani et al., 2018; Elbayad et al., 2019; Schuster et al., 2022). Motivations for this include: (1) different inputs require different amounts of iteration to make a prediction, (2) reducing computation cost.

UT implements a similar mechanism, but the UT version of halting is difficult to interpret. Here we choose a principled version of the dynamic halting mechanism based on the stick-breaking process, viewing it as a probability distribution. First, $\hat{\alpha}_l^{(t)}$ are the halting probabilities predicted by $\text{halt}(\mathbf{h}_l^{(t)})$

**Algorithm 1** Halting mechanism at a given timestep $t$

> **for** $l = 1$ **to** $L$ **do**
>   **if** $\sum_{l'=1}^{l-1} \alpha_{l'}^{(t)} < \alpha_{\text{thresh}}$ **then**
>     $\hat{\alpha}_{l-1}^{(t)} = \text{halt}(\mathbf{h}_{l-1}^{(t)})$
>     $\alpha_{l-1}^{(t)} = \hat{\alpha}_{l-1}^{(t)} \prod_{l'=1}^{l-2} (1 - \hat{\alpha}_{l'}^{(t)})$
>     $\mathbf{a}_l^{(t)} = \text{Attention}(\underbrace{\mathbf{h}_{l-1}^{(t)}}_{Q}, \underbrace{\mathbf{S}_{l-1}}_{K}, \underbrace{\mathbf{S}_{l-1}}_{V})$
>     $\mathbf{h}_l^{(t)} = \text{FeedForward}(\mathbf{h}_{l-1}^{(t)}, \mathbf{a}_l^{(t)})$
>     $\mathbf{s}_l^{(t)} = \left(1 - \sum_{l'=1}^{l-1} \alpha_{l'}^{(t)}\right) \cdot \mathbf{h}_l^{(t)}$
>         $+ \left(\sum_{l'=1}^{l-1} \alpha_{l'}^{(t)} \cdot \mathbf{h}_{l'}^{(t)}\right)$
>   **else**
>     $\mathbf{h}_l^{(t)} = \mathbf{h}_{l-1}^{(t)}$
>     $\mathbf{s}_l^{(t)} = \mathbf{s}_{l-1}^{(t)}$
>   **end if**
> **end for**

, a function which is implemented by an MLP that takes in the previous layer's embedding. Then, the probability of any layer halting is computed by

$$\alpha_l^{(t)} = \hat{\alpha}_l^{(t)} \prod_{l'=1}^{l-1} (1 - \hat{\alpha}_{l'}^{(t)}). \qquad (2)$$

A similar formulation is described in Graves (2016) and Tan and Sim (2016). Algorithm 1 shows how the mechanism is implemented at any given timestep. $\mathbf{h}_{l-1}$ is the output of the previous layer for the current timestep.

Conditioned on the fact that we are computing $\mathbf{h}_l^{(t)}$, time-step $t$ must not have halted before or at $l-1$. So we can use $\mathbf{h}_l^{(t)}$, the unhalted state, as

input to the computation of the attention query of the block. However, since time-step $t$ can attend to all other timesteps, and it these other steps may have halted, we use the halted states $\mathbf{S}_{l-1}$ for the previous layers.

However, because the halting is a 'soft' decision, we can relax the requirement for evaluating all possible halted states and use the expected halted state as a substitute. Previous halting mechanisms use a 'gating' mechanism of convex sums between previously gated outputs and the current step's output $\mathbf{h}_l = \alpha_l \cdot \hat{\mathbf{h}}_l + (1 - \alpha_l) \cdot \mathbf{h}_{l-1}$ (Dehghani et al., 2018). This can lead to vanishingly small gradients going up the layers as $(1 - \alpha_l)$ multiplies. We can instead compute the expected halted embedding at any $l$,

$$\mathbf{s}_l^{(t)} = \underbrace{\left(1 - \sum_{l'=1}^{l-1} \alpha_{l'}^{(t)}\right) \cdot \mathbf{h}_l^{(t)}}_{\text{previous layer if not halted}} + \underbrace{\sum_{l'=1}^{l-1} \alpha_{l'}^{(t)} \, \mathbf{h}_{l'}^{(t)}}_{\text{halted at} < l} \quad (3)$$

If $\alpha_l^{(t)} = 1$ for some $l$, $\mathbf{s}_l^{(t)} = \mathbf{h}_l^{(t)}$, recovering the behavior of the discrete halting decision. We use $\mathbf{s}_l^{(t)}$ as input to the attention key and value transformations.

This probabilistic interpretation also allows us to impose a loss on the expected number of layers used at each step, biasing the model towards fewer iterations, thereby saving computational cost.

$$\mathcal{L}_{\text{ACT}} = \frac{1}{T} \sum_{t=1}^{T} \sum_{l=1}^{L} \alpha_l^{(t)} \cdot l. \quad (4)$$

We use a threshold $\alpha_{\text{thresh}} = 0.999$, such that the cumulative sum of the halting probabilities has exceeded this, no computation will be performed for that time step, and the previous layer's embeddings will be copied. Due to the routing operation required in the implementation fo SMoEs, we can simply route halted states to a "No Op" expert, leading to real savings in computation cost when halting hits the threshold early. We find that adjusting this threshold *after* training can maintain performance while saving computation steps.

## 4 Experiments

First, we show that we can scale the UT with SUT on the WMT'14 English-German (Bojar et al., 2014) translation task. We then ran experiments on Compositional Freebase Questions (CFQ; Keysers et al. 2019) to test for compositional generalization

Table 1: BLEU score on WMT14 En-De translation datasets. MACs (Multiply–Accumulate Operations)[1] measures the computational complexity of each model. [a]Vaswani et al. (2017), [b]Liu et al. (2020), [c]Peng et al. (2020), [d]Zhang et al. (2022), [e]Dehghani et al. (2018), [f]Myle et al. (2018), [g]Wu et al. (2018)

| Model | #Params | BLEU | MACs[1] |
|---|---|---|---|
| Transformer base[a] | 65M | 27.3 | 604M |
| Admin 6L-6L[b] | 61M | 27.7 | 604M |
| MAE-7 [c] | 63M | 28.4 | - |
| MoA base[d] | 65M | 28.4 | 628M |
| UT[e] | 65M | 28.9 | - |
| UT base + SB halting | 64M | 29.3 | 1998M |
| SUT base | 66M | 29.2 | 787M |
| Transformer big[f] | 210M | 29.3 | 2090M |
| LightConv[g] | 202M | 28.9 | 1750M[2] |
| DynamicConv[g] | 213M | 29.7 | 1790M[2] |
| Admin 18L-18L[h] | 151M | 29.0 | 1490M |
| Admin 60L-12L[i] | 256M | 30.1 | 2550M |
| MoA big[d] | 200M | 29.4 | 1220M |
| UT big + SB halting | 105M | 29.6 | 3707M |
| SUT big | 110M | 29.4 | 787M |

Table 2: Ablation Study. "– MIM loss" means replacing the MIM loss with the load balancing loss used in (Fedus et al., 2021). "– MoMHA" means replacing MoMHA with the MoA introduced in (Zhang et al., 2022).

| Model | Valid loss | BLEU | #Params |
|---|---|---|---|
| SUT base | 2.192 | 29.2 | 66M |
| – MIM loss | 2.221 | 28.9 | 66M |
| – MoMHA | 2.232 | 28.7 | 66M |
| – ACT loss | 2.217 | 29.0 | 66M |
| – halting | 2.219 | 29.1 | 65M |

properties. To further analyze the behaviour of the model under compositional generalization settings, we test our model on the Logical inference task from (Bowman et al., 2015). All experiments were implemented within the Fairseq framework (Ott et al., 2019) [2].

### 4.1 English to German Translation

We perform experiments on the WMT'14 English-German translation dataset (Bojar et al., 2014). We use the pre-processing from Liu et al. (2020). We use a joined dictionary and share all word embeddings of the encoder and decoder. For evaluation, we average the last 5 best models according to their negative log-likelihood scores. We report the BLEU scores (Papineni et al., 2002), and also report the MACs (Multiply-Accumulate Operations) to evaluate the runtime computational costs of the

---

[2]Fairseq-based implementation available on: `https://github.com/shawntan/sut`

Table 3: FFD Expert-Word co-occurrences.

| Exp. | 6 | 17 | 41 | 46 |
|------|------|------|------|------|
| Top 5 | a | he | ed | team |
| | their | they | ing | children |
| | his | his | ted | police |
| | this | He | y | devices |
| | an | you | red | system |
| | Det. | Pronouns | Suffixes | Nouns |

different models. MACs of previous models were computed in Zhang et al. (2022).

The results are reported in Table 1. We compare against strong baselines while accounting for the number of parameters in these models. In addition, we train two UTs by setting $E = 1, k = 1$, and parameterizing the FFD and Attention layers with parameters to match our ∼65M, and ∼110M setting for SUT. The SUTs and UTs both demonstrate good parameter efficiency when compared to previous models. In the ∼110M parameter class, SUT and UT perform at around 29.4 and 29.6 BLEU respectively, while previous models require ∼200M parameters. While the SUT does not perform as well as the UT, but the computations required during runtime could be as low as one-fifth of UT. Also, because we keep $k$ constant for SUT, the MACs stays constant as SUT scales up.

We ran experiments removing different aspects of the model and its training process, including: MIM auxiliary loss, Mixture of MHA, the ACT loss, and the halting mechanism. The results are in Table 2. The introduction of multiple heads to the MoA was crucial in seeing performance gains on this task, as well as having the MIM loss as a load-balancing auxiliary objective. Interestingly, halting does contribute as much of a performance gain as it does in CFQ.

Additionally, we compute the top 5 tokens that occur in conjunction with each expert, regardless of layers, and find that certain associations exist. We pick several experts in Table 3 that show a clear sign of co-occurring with tokens that seem to show a pattern. This suggests that while there may be redundancy between the experts, groups of experts can specialize on certain tasks, resulting in some modularity. Future work can investigate if such

---

1The open-source tool PTFLOPS (https://github.com/sovrasov/flops-counter.pytorch) is used to calculate the MACs.

2The MACs values of DynamicConv and LightConv are underestimated. Because the PTFLOPS does not support the customized convolution layers.

modularity can result in more robust generalization.

## 4.2 Compositional Freebase Questions

We run experiments on the Compositional Freebase Questions (CFQ; Keysers et al. 2019) dataset to determine the compositional generalization abilities of the SUT. This is a translation task from natural language to a SPARQL query. As an example, the sequence `Who wrote M1 and wrote a film` would be translated to the target sequence `SELECT DISTINCT ?x0 WHERE { ?x0 a people.person . ?x0 film.writer.film ?x1 M1 . ?x1 a film.film }`. CFQ tests for compositional generalization using the notion of *compound divergence*, which measures how different the training set and test set are in terms of combinations of tokens, which they refer to as compounds. To our knowledge, the current best-performing models either finetune a pretrained language model or, use knowledge about the task to design a suitable prompt for a large language model (Drozdov et al., 2022). While the prompting approach is extremely effective at the CFQ task, we view the task as a benchmark for compositional generalization in general and should be viewed in concert with other experiments, especially real-world data (like translation). When using domain knowledge of the task in the prompt, the results may indicate better performance with a specific approach for CFQ (and perhaps other SQL translation tasks) but might be difficult to extrapolate to other settings.

In our experiments, we use preprocessing scripts from Zheng and Lapata (2021). The scripts perform preprocessing to the target sequence that simplifies the target sequence the same way performed in Furrer et al. (2020). Accordingly, we train a baseline Transformer on the transformed target. We performed a search on the SUT hyperparameters, using the MCD1 validation set, and the best-performing set of parameters are Attention ($E = 1, k = 1, H = 8, D = 64, W = 1$) and FFD ($E = 1, k = 1, D = 1024$), which corresponds to the UT setting. Refer to Appendix A for further details. Since CFQ is a relatively small task, larger scale is not a factor and might suggest that expert specialization may not be as helpful. The results are shown in Table 4. In cases with and without halting, the model already outperforms previous benchmarks, including the UT baseline from Bergen et al. (2021). For a fairer comparison, we use the same hyperparameters as our UT imple-

Table 4: CFQ Results. Results on UT are an average of 5 runs on different seeds.

| Model | Pretraining | MCD1 | MCD2 | MCD3 | Avg. | MACs[1] |
|---|---|---|---|---|---|---|
| T5-based UT (Bergen et al., 2021) | ✗ | 42.7 | 9.5 | 11.6 | 21.3 | 1154M |
| Edge Transformer (Bergen et al., 2021) | ✗ | 47.7 | 13.1 | 13.2 | 24.7 | 6504M |
| Transformer (Keysers et al., 2019) | ✗ | 42.5 | 11.2 | 10.6 | 21.4 | 1154M |
| T5 (Furrer et al., 2020) | ✓ | 61.6 | 31.3 | 33.3 | 42.1 | 1154M |
| Roberta (Zheng and Lapata, 2021) | ✓ | 60.6 | 33.6 | 36.0 | 43.4 | 1660M |
| Dangle (Zheng and Lapata, 2021) | ✓ | 78.3 | 59.5 | 60.4 | 66.1 | 51033M |
| T5-based UT (ours) | ✗ | 68.3 ± 2.9 | 43.1 ± 1.5 | 45.7 ± 1.8 | 52.3 ± 1.6 | 441M |
| UT w/o halting | ✗ | 71.0 ± 3.5 | 48.6 ± 2.3 | 51.3 ± 0.2 | 56.9 ± 1.5 | 654M |
| UT with halting | ✗ | 72.4 ± 3.5 | 51.1 ± 1.8 | 51.7 ± 2.3 | 58.4 ± 1.2 | 654M |

Table 5: Test accuracy of the models, trained on operation lengths of $\leq 6$, with their out-of-distribution results shown here (lengths 7-12). LSTM baseline from Bowman et al. (2015), and Transformer baseline from Shen et al. (2019)

| Model | Number of Operations | | | | | | Comp. Gen. | | |
|---|---|---|---|---|---|---|---|---|---|
| | 7 | 8 | 9 | 10 | 11 | 12 | A | B | C |
| LSTM | 88 | 84 | 80 | 78 | 71 | 69 | 80 | 60 | **59** |
| Transformer | 51 | 52 | 51 | 51 | 51 | 48 | 53 | 51 | 51 |
| SUT | **98** | **97** | **94** | **90** | **88** | **81** | **97** | **94** | 52 |

mentation, we modify our UT implementation to be more similar to the T5-based UT in Bergen et al. (2021). These changes include: the bucketed relative position bias used by T5, and going from post layer-norm to pre layer-norm. While this results in much improved results compared to the original paper, our implementation of UT still outperforms it.

The Dangle (Zheng and Lapata, 2021) model, which beats our model, also requires re-running the encoder for every token decoded. This is an expensive process, but given that both our method and Dangle perform well at this task, is additional evidence that iterative processes are beneficial for compositional generalization.

### 4.3 Logical Inference

We use the logical inference task from (Bowman et al., 2015) as a test bench for UT. Despite the apparent simplicity of the language, the task inherently requires the model to learn the hierarchical structure of the problem. Each instance of the task comprises of two logical statements, and the goal is to predict if the statements are equivalent, contradictory, disjoint, or entail in either direction. For example, given $s_1$ = a and $s_2$ = a ( or b ),

then $s_1 \sqsubset s_2$. The crux of the task is in training the model on sequences that have 0-6 logical operators and evaluating it on sequences that have 7-12 operators. Given our sequence-to-sequence setting, we convert the task into a translation task. The model takes sentence1 #SEP# sentence2 as its source sentence, with the target sentence being the single-token label for that pair.

We train a 12 layer model with Attention ($E = 12, k = 4, H = 2, D = 32, W = 1$) and FFD ($E = 12, K = 4, D = 128$) and halting. Refer to Appendix A for further details. Training a 12-layer Vanilla Transformer achieves approximately the same results as in Shen et al. (2019), so we report their results. Our results in Table 5 confirm the findings of Tran et al. (2018), showing that with recurrence in SUTs, we are able to generalize to longer sequences of the task. While there are other models that induce a tree structure that performs exceedingly well on the task, we wanted to evaluate our model against other popular architectures. The LSTM is a strong baseline, and we find that UT outperforms it in generalization. We also evaluate UTs on the compositional generalization splits as proposed in (Shen et al., 2019), where the splits A, B, and C are in increasing difficulty. The results show that UTs are able to generalize better for the A and B splits, outperforming the LSTM and VT. Split C is still presents a challenge for the Transformer variants.

Additionally, we compute the average halting depth for the test data segmented by operator counts. Because more operators require more nesting of expressions in the sequences, more recursion is required to properly parse the sequence. As expected, in Figure 4, the average halting depth increases as more operators are used. The operator count for these clauses are correlated with length,

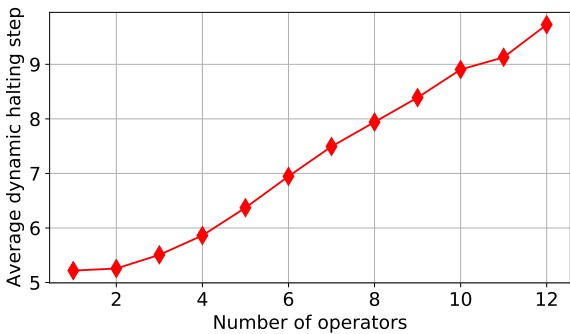

Figure 4: The average dynamic halting depth of the UT model as the number of operators increases in the test set. The model learns to think more when the problem is harder.

which suggests that SUTs may be suited to generalize for length. We include further experiments on length generalization in the Appendix Table 8.

## 4.4 Post-training Computation Reduction

Does lowering $\alpha_{\text{thresh}}$ *after* training cause the model to halt earlier, saving computation? How much would that cost us in terms of accuracy?

We estimate the skipped SUT block computations given different values of $\alpha_{\text{thresh}} \in \{0.1, 0.2, \ldots, 0.9\}$ by looking at the halting patterns of the decoder given the ground truth source-target pairs. We pass the source-target pair into the model and analyze the halting patterns of the model, giving us a rough estimate of how much computation would be saved as a percentage of computing all layers of the SUT.

**Logical Inference** We observe the resulting performance on the hardest split of the test set with 12 operations. Due to the already saturated halting pattern, the halting probability $\alpha_l$ spikes rapidly from close to 0 to higher values, resulting in a near constant $\sim 50\%$ reduction of the computation time regardless of the threshold.

**CFQ** Using the MCD1 test split of the dataset, and our best-performing model on MCD1, we perform the $\alpha_{\text{thresh}}$ adjustment. The halting patterns reflect the repeated structure of SQL, using fewer steps for '.' and 'WHERE', while the main bulk of the region within {...} requires more SUT steps before halting. Surprisingly, when $0.8 \leq \alpha_{\text{thresh}} \leq 0.999$, the accuracy remains fairly constant. An estimated 33% of the computation steps were skipped at $\alpha_{\text{thresh}} = 0.8$. At $\alpha_{\text{thresh}} = 0.1$, there is a slight increase in the number of computed steps, which

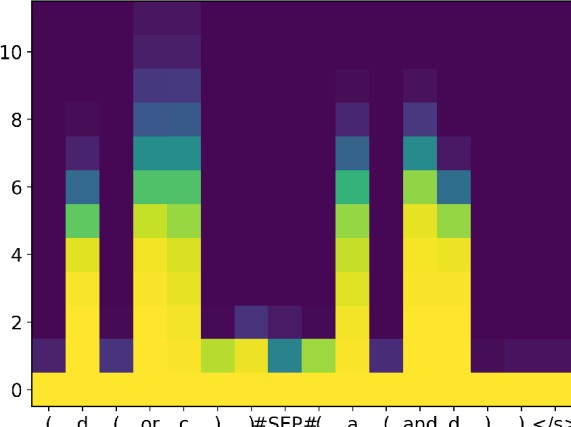

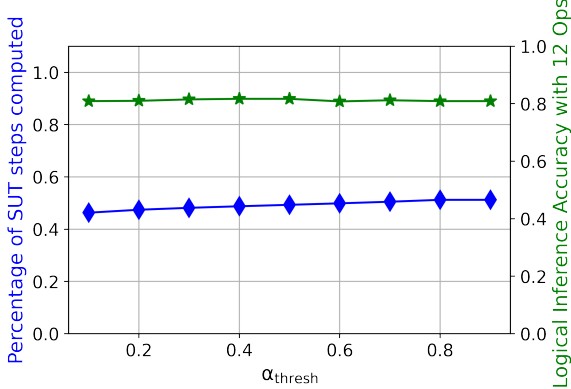

Figure 5: *Above:* Plot of $1 - \sum_{l'=1}^{l-1} \alpha_{l'}^{(t)}$, for an example Logical Inference input — $x$-axis: timesteps, $y$-axis: layers. This visualizes the halting pattern of the model: dark blue represents halted, while yellow represents active. *Below:* Efficiency vs. Performance tradeoff curves when $\alpha_{\text{thresh}}$ is adjusted.

is possible since halting earlier will result in different embeddings, and result in different halting decisions in other timesteps. Overall, the results suggest that we can save about 20% of the SUT computation steps without any drop in accuracy, and about $\sim 50\%$ for a 0.2% decrease.

**English-German Translation** For this larger dataset, we find that these translation models halt much later, suggesting that the translation task requires more computational steps than the 6-layer SUT we used. However, further increasing the number of layers to 12 layers does not bring about much benefit, as evidenced by the halting in Figure 4, which is an example of the halting mechanism using nearly all layers. For comparison, Admin 60L-12L model, requires a 60-layer encoder to achieve its performance. Even when $\alpha_{\text{thresh}} = 1$, the skipped computation steps remain at about 33%, compared to 80% in the CFQ task. We find that we can reduce the computation by 9% while still

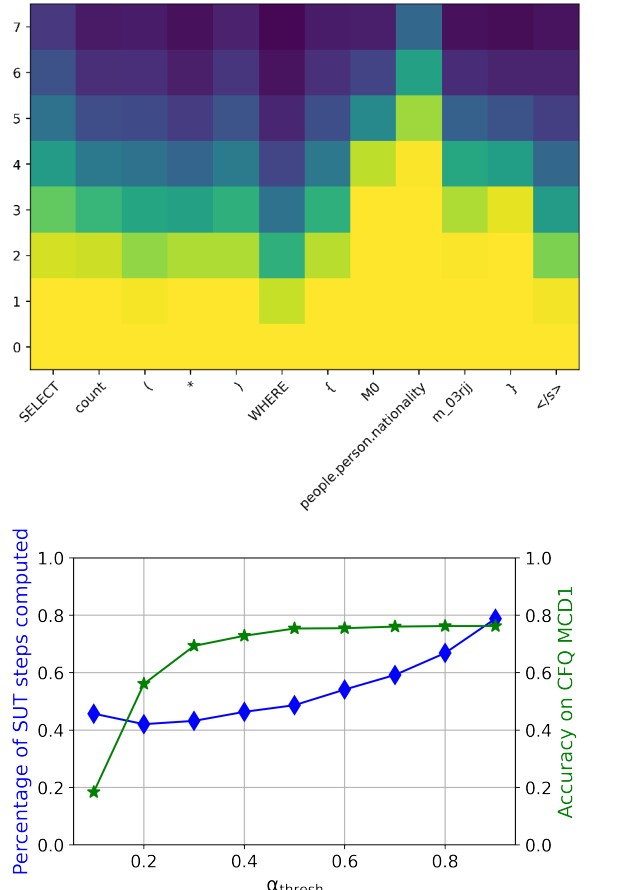

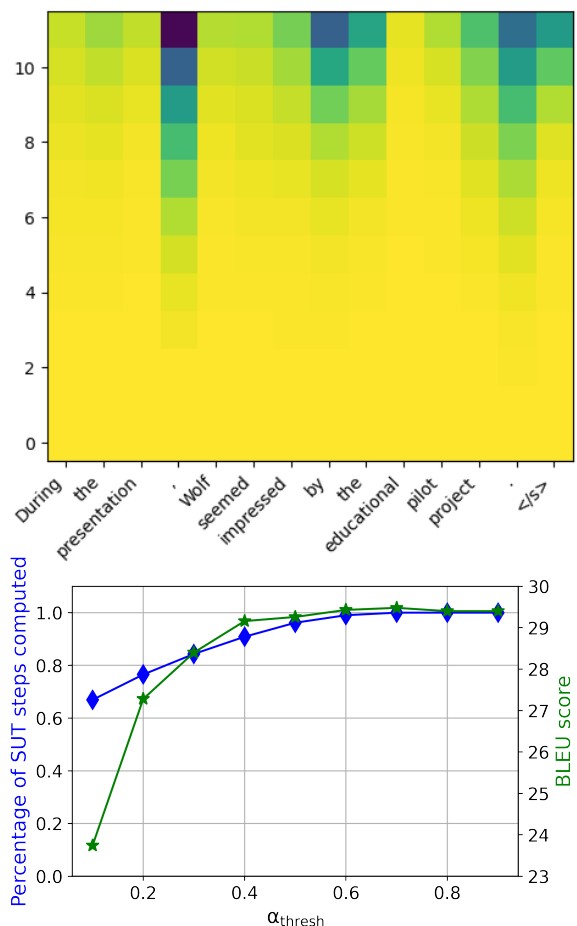

Figure 6: Halting plot and trade-off curves for CFQ. (See Figure 5 for description)

Figure 7: Halting plot and trade-off curves for English-German Translation. (See Figure 5 for description)

retaining a BLEU score of 29.1.

# 5 Conclusion

We show that it is possible to scale up the UT via SUTs, and SUTs outperforms models of the same capacity in the WMT'14 English-to-German translation task. The recursive nature of both UTs and SUTs allows for better inductive biases, which we have demonstrated in synthetic tasks like CFQ and logical inference. VTs have been shown to be poor at these compositional generalization tasks without additional domain knowledge. The stick-breaking dynamic halting mechanism also allows post-training adjustment of computation cost, which is a boon for deployment at scale.

**Limitations** While the experiments in this paper show the desirable generalization properties of UTs, there are some aspects of compositional generalization that SUTs do not solve. Importantly, while we demonstrate scaling UTs up via SMoEs, further experiments on larger settings are needed

to ascertain viability in large scale systems. Other issues may also crop up in the further scaling of SUTs, but we believe there is ample literature to draw on to finding solutions for these problems.

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
