# OpenReview forum: "Sparse Universal Transformer"
_EMNLP/2023/Conference — EMNLP 2023 Main_

### Official Review · Reviewer_PDdV · 2023-07-31

**Typos Grammar Style And Presentation Improvements:** 188
**Soundness:** 4

**Excitement:**

3: Ambivalent: It has merits (e.g., it reports state-of-the-art results, the idea is nice), but there are key weaknesses (e.g., it describes incremental work), and it can significantly benefit from another round of revision. However, I won't object to accepting it if my co-reviewers champion it.

**Missing References:**

The correct citation for Zhang et al appears to be: https://aclanthology.org/2022.emnlp-main.278/


**Paper Topic And Main Contributions:**

This paper describes a variant of the Universal Transformer that makes a few modifications:

1. Instead of the original halting condition, which halted at a given position if a certain activation exceeded a certain threshold, the paper proposes a new halting mechanism, in which the halting probabilities (\alpha) are actually probabilities: the forward pass sums (approximately) over all possible outcomes of the halting decisions. This halting mechanism appears to be fairly helpful (+0.4 BLEU).

2. Sparse mixture of experts (Shazeer et al., 2017) for both the attention (Zhang et al, 2022; Shen et al, 2023) and feedforward layers. This hurts accuracy only slightly, but makes the network several times faster.




**Questions For The Authors:**

A. Not sure if this was addressed in the original UT paper, but how important is it to have the halting decision made per-position? How different would it be to have one global halting decision?

B. Are the use of sparsity and the new halting mechanism orthogonal improvements to the model, or is there a connection between the two?


**Reasons To Accept:**

Halting seems to be the trickiest part of designing/implementing a Universal Transformer, and this paper presents a nicer mechanism that seems to work better. I consider this the main contribution of the paper and would even suggest changing the title of the paper.

The adaptation of previous work on SMoE to the Universal Transformer to provide a significant speedup is also a good contribution of the paper.


**Reasons To Reject:**

Unless I am mistaken, SMoE and the MIM loss are both previous work, and the novelty in this part of the paper lies only in applying them to the Universal Transformer.

The description of the method is not clear and/or relies too much on formulations in previous papers (see below under "Presentation Improvements"). Some more background, at minimum definitions of all the variable names used, is needed.


**Reproducibility:**

4: Could mostly reproduce the results, but there may be some variation because of sample variance or minor variations in their interpretation of the protocol or method.

**Reviewer Confidence:**

4: Quite sure. I tried to check the important points carefully. It's unlikely, though conceivable, that I missed something that should affect my ratings.

---

> ### Author Rebuttal · Authors · 2023-08-28
>
> Thank you for your in-depth review.
>
>
> > A. Not sure if this was addressed in the original UT paper, but how important is it to have the halting decision made per-position? How different would it be to have one global halting decision?
>
> In the cases that we’ve seen where the SUT does halt early, the tokens are usually tokens that are less important: EOS, punctuation, stop-words, etc., and uses more compute where needed.
> Considering this, we think that if the goal of early halting is to save computation costs, then allowing the model to reduce computations on tokens where it is not necessary would help to achieve this.
>
> > B. Are the use of sparsity and the new halting mechanism orthogonal improvements to the model, or is there a connection between the two?
>
> Each can be implemented without the other.
>
> However, there is a connection between them that arises out of convenience. In order to perform the sparse expert computation, we have to route each hidden state to the associated expert. Without this, the computational savings from sparse computation cannot be realised. In doing this, we can assign halted states to a “no-op” expert, where they simply pass-through to the output, thereby allowing the computation savings on halted states.
>
> The alternative without this specific implementation detail would be to evaluate each hidden state through all the available experts, selecting the top-$k$ experts, and then replacing the halted states with the previous output, which creates a lot of redundant computation steps.
>
>
> Thank you for pointing out these missing definitions, we will include these definitions in the paper. Here are their definitions to clarify things:
>
> > 229: what is \mathcal{X}?
>
> It is the set of all the hidden states for all timesteps at all layers.
>
>
> > 188: what’s the definition of g?
>
> This is the marginal probability of the experts over all the routing probabilities, and we assume a uniform probability over all time-steps:
> $$g(e) = \frac{1}{|\mathcal{X}|} \sum_{\mathbf{h} \in \mathcal{X}} g(e|\mathbf{h}) $$
>
>
>
> > 228: what’s the intuition for \mathcal{L}_{MIM}?
>
> The entropy term (Line 228) increases the entropy of the marginal probability of the router predictions, which at its maximum means that the weight for each router across the entire minibatch is uniform.
> The conditional entropy term (Line 229) decreases the conditional entropy, which causes the prediction of the router to be sharp, and also penalises the uniform distribution solution for the router.
>
>
> > 234: what’s the router?
>
> We apologise for the confusion. We use the term gating network and router interchangeably, and we should make sure the terminology is consistent throughout the paper. We will make sure to keep consistency.
>
> > page 4: what’s \mathbf{s}?
>
> It is the weighted halted state as defined by Line 280.
>
> > 275: not very clear. It might be helpful to write out the full distribution over all 2^T possible combinations of halted/not halted, and then write \mathbf{s} as an expectation under this distribution, pointing out that the expectation can be computed efficiently.
>
> This is a great suggestion, we will express it this way to make things clearer.
>
> We hope we've addressed your concerns for this paper.

---

### Official Review · Reviewer_EnEq · 2023-08-02

**Soundness:** 5

**Excitement:**

4: Strong: This paper deepens the understanding of some phenomenon or lowers the barriers to an existing research direction.

**Paper Topic And Main Contributions:**

This paper proposes a more computationally efficient version of the Universal Transformer (i.e. a transformer with a variable number of layers) by applying the method of Sparse Mixture of Experts (SMoE) to the feedforward and multi-head attention mechanisms. SMoE allows the model to evaluate only $k$ of $E$ submodules in a network, where $k < E$, saving computation. The authors also propose a simpler dynamic halting mechanism than the one used in the original Universal Transformer. They call their model Sparse Universal Transformer (SUT).

The main experimental results are:
1. MT experiments on WMT 14 English-German. In some cases, SUT has better translation quality, lower computational cost, and fewer parameters than baseline models.
2. Experiments on the Compositional Freebase Questions task, meant to test compositional generalization. SUT does better than all previous baselines, except one that uses a pretrained LM.
3. Experiments on the logical inference task of Bowman et al. (2015), where the test set includes out-of-distribution/longer examples. SUT outperforms baseline LSTMs and Transformers, except on a difficult subset of the test set where LSTMs are still best. Fig 4 shows that the number of layers SUT uses increases with the number of operators in the example, as expected.
4. Analysis of the tradeoff between stopping computation early and performance, for multiple tasks.

In summary, this paper proposes a new Universal Transformer architecture, SUT, that outperforms a wide range of baselines on multiple tasks, including ones that test compositional generalization, while decreasing computational cost compared to standard UT.


**Questions For The Authors:**

A. 288: What is $g(e)$?

B. Do the computational savings for SMoE only apply to inference, not training?

C. Tab 3: This is a cool result. How much cherry-picking did you have to do to get lists like these? Are there lots of obvious groupings like this, or are most of the experts uninterpretable?

D. Tab 4: What is the metric being used here? Is higher or lower better?

E. Tab 4: Are all UT models sparse in this table?

F. Tab 5: What is the metric here?

G. Why is it called "stick-breaking"?

H. Tab 4: Is "UT with halting" your full SUT model?


**Reasons To Accept:**

Overall, the paper is very well-written and well-motivated, and I consider the method to be a creative and novel combination of existing ideas. The experimental methodology appears sound throughout. The introduction and background sections are excellent at motivating the method and explaining the main ideas behind it. The breadth of the related work discussed and baselines tested against is comprehensive.

I found Fig 4 to be a particularly cool result. Table 3 is also interesting.


**Reasons To Reject:**

In Table 1, the bottom half seems to support the authors' story, whereas the top half does not. For the ~66M tier, SUT has slightly higher BLEU than the baselines, but it also has more params and MACs. I'm not sure what conclusions to draw from this, if any, and I would appreciate more discussion of this by the authors.

In Tab 1 and Tab 2, a lot of the BLEU scores are very close, so it would be helpful to see std devs or significance testing. Otherwise, it's a bit hard to be confident in some of these results.


**Reproducibility:**

4: Could mostly reproduce the results, but there may be some variation because of sample variance or minor variations in their interpretation of the protocol or method.

**Reviewer Confidence:**

3: Pretty sure, but there's a chance I missed something. Although I have a good feel for this area in general, I did not carefully check the paper's details, e.g., the math, experimental design, or novelty.

**Typos Grammar Style And Presentation Improvements:**

A. 209: K -> k

B. 266: extra "it"

C. 310: Use \citet

D. 338: extra "but"

E. 369: Can you explain what this syntax means?

F. 423: missing subject

G. 464: Extra "is"

H. Fig 5: Can you provide a legend for the heatmap colors? Maybe black and white would be simpler and easier to read.

I. Fig 5: The text is too small to read.

J. Fig 5: Is this figure not referenced in the text?

K. 502: Wrong quotes

---

> ### Author Rebuttal · Authors · 2023-08-28
>
> Thank you for the detailed review of our paper.
>
> > In Table 1, the bottom half seems to support the authors' story, whereas the top half does not. For the ~66M tier, SUT has slightly higher BLEU than the baselines, but it also has more params and MACs. I'm not sure what conclusions to draw from this, if any, and I would appreciate more discussion of this by the authors.
>
> Many of the hyperparameters used in our models were derived from the settings used in the Mixture of Attention Heads (Zhang et. al., 2022) paper, with two additions:
> 1. Increasing the number of experts since there was only one Transformer block now, and
> 2. An MLP router router for both the Attention and the MLP module in the Transformer block.
>
> These additions lead to one more router computation for every iteration performed by the SUT (higher MACs), and a wider output layer for the router due to the extra experts and additional router (overhead in the number of parameters).
>
> We now have results on a smaller model: 43M parameters, 28.4 BLEU, and on plotting these three settings (BLEU vs. Parameter count), we find that the SUT models form a Pareto front for the existing methods within the same parameter count.
> We hope this addresses your biggest concern, we will include this plot in the paper.
>
>
> > In Tab 1 and Tab 2, a lot of the BLEU scores are very close, so it would be helpful to see std devs or significance testing. Otherwise, it's a bit hard to be confident in some of these results.
>
> We agree and other reviewers have also suggested this.  We are presently running these additional runs to provide these error bars.
> However, we would like to point out that our goal is to show that SUT can reach comparable results while using fewer parameters and computation costs.
>
>
>
> > A. 288: What is g(e)?
>
> We apologise for the confusion. This is the marginal probability of the experts over all the routing probabilities, and we assume a uniform probability over all time-steps:
> $$g(e) = \frac{1}{|\mathcal{X}|} \sum_{\mathbf{h} \in \mathcal{X}} g(e|\mathbf{h}) $$
> We will include this definition in the paper.
>
> > B. Do the computational savings for SMoE only apply to inference, not training?
>
> It is for both training and inference, this is also the case in [Shazeer et. al. 2017](https://arxiv.org/pdf/1701.06538.pdf).
>
>
> > C. Tab 3: This is a cool result. How much cherry-picking did you have to do to get lists like these? Are there lots of obvious groupings like this, or are most of the experts uninterpretable?
>
> We took the experts that seemed to have an interesting pattern in the top 5 tokens, from among 24 experts. There were others that had an interesting mix of 2 (proper nouns and pronouns) for example. The issue is that it’s hard to evaluate what the experts have specialised in after training because at different levels of the SUT, they may be performing different operations which may not correspond to the word at that particular time-step, whereas in this particular table, we have chosen to analyse the co-occurences of expert usage and words.
>
> We will include the full list of top words for each expert in the appendix.
>
> > D. Tab 4: What is the metric being used here? Is higher or lower better?
> > F. Tab 5: What is the metric here?
>
> We apologise, these are Accuracy scores. In Tab. 4 they are results for a full match on the SQL target sequence, so higher is better.  In Tab. 5, the results are how many of the logical expressions were classified correctly, again, higher is better. We will make this clear in the tables.
>
> > E. Tab 4: Are all UT models sparse in this table?
>
> In this table all UT models are dense.
>
> As suggested in the WMT’14 En-De experiments, the results show that when we are able to use a dense UT of equivalent parameters to its SUT counterpart, the dense UT performs better, but at great computation cost.
> We mention performing a hyperparameter search on MCD1 and its validation set, and finding that $E=1$ was the best setting for CFQ.
>
> Since CFQ is a small task and requires only a small model, the benefits of sparsity do not play a role in this experiment. However, we wanted to demonstrate that the UT setting (along with the additional updates we introduced in this paper) still confers some compositional generalisation properties.
>
>
> > G. Why is it called "stick-breaking"?
>
> It is an alternative formulation of the Dirichlet process that formulates it as sampling from a series of Bernoulli random variables till success,
>
> $$\beta_{k}=\beta '_{k} \prod _{i=1}^{k-1} (1 - \beta'_i)$$
>
> where $\beta'_{k} \in (0, 1)$
>
> Consider starting with a unit-length stick and breaking off  $\beta'_k$ portion of the the remaining length of stick.
>
> $\beta_k$ then gives the length of the $k$-th break in relation to the original length.
> Because of this formulation, summing up $\beta_{k}$ approaches 1.
> Further explanation can be found [here](https://en.wikipedia.org/wiki/Dirichlet_process#The_stick-breaking_process).
>
> > H. Tab 4: Is "UT with halting" your full SUT model?
>
> Yes. It is a special case of the SUT model with $E=1$ (1 expert), with the halting mechanism.
>
> We hope this addresses your concerns for the paper, and we would also like to thank you for the detailed list of typos and presentation improvements you suggested. We will make the appropriate modifications to the paper.

---

### Official Review · Reviewer_T1no · 2023-08-04

**Soundness:** 3

**Excitement:**

4: Strong: This paper deepens the understanding of some phenomenon or lowers the barriers to an existing research direction.

**Paper Topic And Main Contributions:**

The paper introduces a new architecture, Sparse Universal Transformer, which is MoE applied to Universal Transformer. The architecture achieves three desirable properties which previously had not been achieved simultaneously: the recurrence of Universal Transformer (using weight tying), high capacity of Transformer, and compute efficiency of Transformer. The architecture is evaluated on three tasks: WMT'14 and two benchmarks for compositional generalization, CFQ and Logical Inference. The paper finds comparable or stronger performance for SUT compared to Transformer and Universal Transformer baselines. The paper also conducts some ablation experiments on specific parts of the architecture.

**Questions For The Authors:**

A) The paper does not provide standard errors its measurements. It is therefore not clear which of the reported differences are real. This is particularly an issue for the ablation results, where many of the differences are small.

**Reasons To Accept:**

I am excited about this paper. The architecture is technically sound and well motivated. The paper finds a solution to a real problem for Universal Transformers, which is poor scaling of compute with parameter count. The proposal is the obvious-in-retrospect solution to this problem, which is a strength given the proposal's novelty in the literature.

The paper is well written, and clearly communicates the details of the architecture, the experiments, and their implications.

**Reasons To Reject:**

The experimental evaluation is thin and not fully convincing. Given the "obviousness" of the architecture (this is not meant pejoratively, rather it is sufficiently natural that other researchers are likely to think of it), I would support publication of the paper regardless of whether SUT is in fact stronger than Transformers or Universal Transformers in any meaningful way. Even if the result is negative, this will save future researchers a great deal of wasted effort. However, I would not recommend publication in its current state. What I would find sufficient is a convincing demonstration of the model's strengths and weaknesses, whatever they are.

- One of the paper's central claims is that SUT improves compositional generalization. However, the paper does not evaluate on most of the major datasets in this area (COGS, SCAN, PCFG). This would be necessary to make a convincing case for improved generalization.

- The only natural dataset that the paper evaluates on is WMT'14. It is therefore unclear how the architecture would perform on natural data. Language modeling could be interesting, or some reasoning tasks given the paper's emphasis, though those tasks would require pretraining.

- Despite discussing scaling, the paper does not estimate scaling laws for the architecture. There have been many Transformer alternatives proposed in the literature, which have been found to have worse scaling behavior than Transformers (Tay et al., Scaling Laws vs Model Architectures: How does Inductive Bias Influence Scaling?). It seems important to know whether SUT falls into this group. This should be feasible since scaling laws can be accurately estimated from sub-billion parameter models.

- The model for CFQ uses only a single expert, in which case it reduces to a normal Universal Transformer. This does not provide evidence for SUT. It is also strange that SUT outperforms Universal Transformer in this setting.

**Reproducibility:**

4: Could mostly reproduce the results, but there may be some variation because of sample variance or minor variations in their interpretation of the protocol or method.

**Reviewer Confidence:**

4: Quite sure. I tried to check the important points carefully. It's unlikely, though conceivable, that I missed something that should affect my ratings.

---

> ### Author Rebuttal · Authors · 2023-08-28
>
> Thank you for your in-depth review of our paper!
>
> > One of the paper's central claims is that SUT improves compositional generalization. However, the paper does not evaluate on most of the major datasets in this area (COGS, SCAN, PCFG). This would be necessary to make a convincing case for improved generalization.
>
> By your recommendation, we ran some experiments, and here are some preliminary results for SCAN and PCFG productivity & systematicity splits.
>
>
> |         Task         |      Vanilla Transformer     | Vanilla Transformer (ours) | UT (SUT 1 expert) |
> |----------------------|------------------------------|----------------------------|-------------------|
> | SCAN (add_turn_left) | 99.6% (Furrer et. al., 2020) |                        20% |              100% |
> | SCAN (add_jump)      | 1% (Furrer et. al., 2020)    |                       7.3% |                0% |
> | SCAN (length)        | 0% (Furrer et. al., 2020)    |                         0% |                7% |
> | PCFG (systematicity) | 72% (Hupkes et. al., 2019)   |                        76% |               87% |
> | PCFG (productivity)  | 52% (Hupkes et. al., 2019)   |                        56% |               62% |
>
>
> In all cases, we ran a baseline Vanilla Transformer (VT) model using the IWSLT Transformer model from fairseq, and compared with a relevant VT baseline from the literature.
> We were unable to replicate the `add_turn_left` result in Furrer et. al. on SCAN, and found that our VT results on the PCFG splits were slightly higher.
>
>
> These results show that there are nuances to compositional generalisation. While the SUT can bring consistent benefits in the case of PCFG, the harder generalisation aspects of `add_jump` and `length` in SCAN may require a different kind of generalisation.
> To our knowledge, there is work that deals explicitly with the `add_jump` ([Russin et. al., 2019](https://aclanthology.org/D19-1438.pdf), [Li et. al., 2019](https://aclanthology.org/D19-1438.pdf)) and others that deal with the `length` split by decoding in a tree structure ([Tan et. al. 2020](https://aclanthology.org/2020.findings-emnlp.208.pdf), [Yoon, 2021](https://arxiv.org/abs/2109.01135)).
> SUT tackles some of these aspects of compositional generalisation, but not all, and this is a good illustration of its strengths and weaknesses, and we will add this to the paper. Thank you for the suggestion.
>
> We also have Long-range Arena ListOps results in the appendix, where we also achieve good long-range performance.
>
> > The only natural dataset that the paper evaluates on is WMT'14. It is therefore unclear how the architecture would perform on natural data. Language modeling could be interesting, or some reasoning tasks given the paper's emphasis, though those tasks would require pretraining.
>
> We are currently working towards large language modeling pre-training with the SUT.
>
> >Despite discussing scaling, the paper does not estimate scaling laws for the architecture. There have been many Transformer alternatives proposed in the literature, which have been found to have worse scaling behavior than Transformers (Tay et al., Scaling Laws vs Model Architectures: How does Inductive Bias Influence Scaling?). It seems important to know whether SUT falls into this group. This should be feasible since scaling laws can be accurately estimated from sub-billion parameter models.
>
> We believe that estimating scaling laws requires a more diverse set of tasks at many different scales.
> However, we specifically targeted issues with scaling UTs by introducing sparsity into the UT block. In the appendix of Kaplan et. al., 2020 (Page 24, Figure 17), the authors do note that the universal transformer has better scaling properties in terms of parameter efficiency, but when compared in terms of computation, it performs worse. Scaling Laws vs Model Architectures (Tay et. al., 2022, Appendix 4.1) mentions the same difficulty with scaling UTs, stating: "we found that even with an increase of FLOPS, this does not lead to improved performance".
> We do provide an explanation in our paper for this (Figure 1), namely that attempting to reach parameter equivalence with a VT for a standard UT will incur a quadratic scaling relathionship.
> Sparsifying the UT directly tackles the issue.
>
>
> > The model for CFQ uses only a single expert, in which case it reduces to a normal Universal Transformer. This does not provide evidence for SUT.
>
> CFQ is a small task that is not knowledge-intensive, and thus requires a smaller model overall, which requires far fewer experts.
> The model used in the CFQ task is a special instantiation of the SUT with $E=1$, after performing a hyperparameter search on MCD1 and its validation set.
> Ultimately, the issue of overfitting with higher parameters still plays a part in the final result.
>
> We also wanted to demonstrate that the UT setting (along with the other additions we introduced in this paper) still confers some compositional generalisation properties.
>
> > It is also strange that SUT outperforms Universal Transformer in this setting.
>
> The "Universal Transformer" setting varies slightly from paper to paper. In several cases, as in Bergen et. al. 2021, authors will refer to a model as a Universal Transformer when the weights of a given Transformer architecture are tied across layers. However, we find that certain architecture choices do affect performance on CFQ:
> In the case of T5,
> 1. The choice of pre-LayerNorm over post-LayerNorm, (we believe this affects gradient propagation in a recurrent model),
> 2. The type of positional embeddings used (T5 uses binned positional biases instead of an embedding),
> 3. The size of the model (the Bergen et. al. 2021 model is much larger),
>
> We attempt to account for these discrepancies in our T5-based UT baseline and reported those results, which fared much closer to our variant of UT.
> We have mentioned this in the paper but will elaborate further to clarify the point.
>
> > A) The paper does not provide standard errors its measurements. It is therefore not clear which of the reported differences are real. This is particularly an issue for the ablation results, where many of the differences are small.
>
>
> We agree and other reviewers have also suggested this. We are presently running these additional runs for WMT'14 En-De to provide these error bars.
> We provide these standard errors for the CFQ tasks, where we report the average of 5 runs, but follow the Zheng and Lapata, 2021 format by reporting the average only.
>
> | Model              | MCD1            | MCD2           | MCD3           |
> |--------------------|-----------------|----------------|----------------|
> | T5-based UT (ours) | 68.3 $\pm$ 2.9  | 43.1 $\pm$ 1.5 | 45.7 $\pm$ 1.8 |
> | UT w/o halting     | 71.0 $\pm$ 3.5  | 48.6 $\pm$ 2.3 | 51.3 $\pm$ 0.2 |
> | UT with halting    | 72.4 $\pm$ 3.5  | 51.1 $\pm$ 1.8 | 51.7 $\pm$ 2.3 |
>
> We hope the additional results and explanations have addressed your concerns, and also hope you will reconsider the score for the soundness of our paper. Thank you!

---

### Meta-Review · Area_Chair_wsg7 · 2023-09-19

**Recommendation:** 5

**Metareview:**

The paper introduces the Sparse Universal Transformer (SUT), an innovative variant of the Universal Transformer that integrates the Sparse Mixture of Experts (SMoE) into the feedforward and multi-head attention mechanisms. This alteration offers computational efficiency and the presented dynamic halting mechanism provides a simpler approach than the original Universal Transformer. Through experiments, SUT demonstrates enhanced translation quality, improved compositional generalization, and better computational trade-offs on several benchmarks like WMT 14 English-German. However, concerns have been raised about the thoroughness of the experimental evaluation, the clarity of method descriptions, and the originality of some contributions.

Pros:

Introduction of a well-motivated and technically sound SUT architecture that amalgamates the efficiency of Transformers with the recurrence of Universal Transformers.

A more efficient dynamic halting mechanism, which not only simplifies but also improves performance.

The paper's comprehensive approach, detailing a breadth of related work and providing results against numerous baselines.

Cons:

Parts of the paper's method, especially concerning the SMoE and MIM loss, seem to rely heavily on prior work, questioning the originality of the contribution.

Insufficient evaluation on some datasets (partially addressed in rebuttal).

---

### Decision · Program_Chairs · 2023-10-07

**Decision:**

Accept-Main

**Comment:**

The paper introduces the Sparse Universal Transformer (SUT), an innovative variant of the Universal Transformer that integrates the Sparse Mixture of Experts (SMoE) into the feedforward and multi-head attention mechanisms. This alteration offers computational efficiency and the presented dynamic halting mechanism provides a simpler approach than the original Universal Transformer. Through experiments, SUT demonstrates enhanced translation quality, improved compositional generalization, and better computational trade-offs on several benchmarks like WMT 14 English-German. However, concerns have been raised about the thoroughness of the experimental evaluation, the clarity of method descriptions, and the originality of some contributions.

Pros:

Introduction of a well-motivated and technically sound SUT architecture that amalgamates the efficiency of Transformers with the recurrence of Universal Transformers.

A more efficient dynamic halting mechanism, which not only simplifies but also improves performance.

The paper's comprehensive approach, detailing a breadth of related work and providing results against numerous baselines.

Cons:

Parts of the paper's method, especially concerning the SMoE and MIM loss, seem to rely heavily on prior work, questioning the originality of the contribution.

Insufficient evaluation on some datasets (partially addressed in rebuttal).